# Gender differences in suicide-related communication of young suicide victims

Elias Balt[1]*, Saskia Mérelle[1], Diana van Bergen[1,2], Renske Gilissen[1], Pommeline van der Post[1], Milou Looijmans[1], Daan Creemers[3], Sanne Rasing[3,4], Wico Mulder[5], Lieke van Domburgh[6,7], Arne Popma[7]

1 Research department, 113 Suicide Prevention, Amsterdam, The Netherlands, 2 Department of Pedagogical and Educational Sciences, Faculty of Behavioral Social Sciences, University of Groningen, Groningen, The Netherlands, 3 Child and Adolescent Psychiatry, GGZ Oost Brabant, Boekel, The Netherlands, 4 Behavioral Science Institute, Radboud University, Nijmegen, The Netherlands, 5 Youth healthcare, Dutch Centre for Youth Health (NCJ), Utrecht, The Netherlands, 6 Quality of Care & Innovation, Pluryn, Nijmegen, The Netherlands, 7 Child and Adolescent Psychiatry & Psychosocial Care, Amsterdam University Medical Centre (AUMC), Amsterdam, The Netherlands

* e.balt@113.nl

## Abstract

### Objectives

There is limited insight into gender differences in suicide-related communication (SRC) in youths. SRC is defined as *"the act of conveying one's own suicide ideation, intent or behaviours to another person"*. Increasing our understanding of SRC in youths will enable us to recognize and specify needs of female versus male youths. The current study explores SRC in a sample of Dutch suicide victims aged under 20 and examines gender differences.

### Methods

Interview data from a psychological autopsy study of 35 youths who died by suicide in the Netherlands in 2017 were analysed. Qualitative analyses were performed to examine *explicit* SRC throughout the youths' lives and *implicit* SRC during the last months prior to suicide. We employed the Constant Comparative Method to explore patterns in the debut, form, frequency, medium, content, type of recipient, and SRC in the last months prior to suicide death.

### Results

We identified commonalities in the SRC of youths, including the content of suicide notes and an emphasis on suicide method and preparation in the last months. Girls, however, had an earlier debut of SRC, a higher frequency of *explicit* SRC, and more often directed SRC towards varied types of recipients compared to boys. Moreover, SRC of girls seemed focused on coping and achieving support from others more than SRC of boys. The SRC of boys in comparison to girls was often ambiguous or diluted by "humorous" connotations.

**Data Availability Statement:** The interview data cannot be shared publicly because of ethical restrictions: the dataset contains potentially identifying and sensitive information. The Medical

Research Ethics Committee (MREC) of Amsterdam UMC has imposed this restriction (registration number: 2018.651 – NL68348.029.18). Data are available upon request from the Ethics Committee for researchers who meet the criteria for access to confidential data. A request may be submitted by mail to metc@vumc.nl.

**Funding:** The author(s) received no specific funding for this work.

**Competing interests:** No authors have competing interests.

## Conclusion

Unique patterns in SRC of boys and girls posed corresponding challenges for next of kin to interpret communications and respond adequately to SRC. The early debut of girls' SRC highlights the importance of early screening and prevention efforts in girls, while the late debut and ambiguity in boys' SRC implores professionals and next of kin to encourage young males to be unequivocal about suicide ideation or intent.

## Introduction

Suicide is a leading cause of death in young individuals in the Netherlands [1] and the third leading cause of death in young people aged 15–29 worldwide [2]. Suicidal behaviours (i.e. suicide ideation, deliberate self-harm, and suicide attempts) often debut between the ages of 12–17 [3, 4]. The suicidal process is the timeframe in which an individual moves from first ideation towards the act of an attempted or completed suicide. The timeframe from ideation to lethal suicide shows large individual variation. Previous research showed that it can be as short as two months in young individuals (aged 15–29) and in 28% of youths spanned three months or less [3]. Approximately one third of adolescents with suicide ideation attempt suicide [5]. Sometimes, the suicidal process can go unnoticed until death by suicide occurs [6].

In what is often referred to as the gender paradox [7], males tend to have 2–4 times higher suicide rates, whereas suicide attempts are 3–9 times more common in females [4, 8]. In the Netherlands, 1.85 per 100,000 boys versus 1.23 per 100,000 girls aged under 20 died by suicide between 2015 and 2019 [9]. Gender differences in the suicidal process manifest during early adolescence [10, 11]. Kaess and colleagues [11] determined lifetime suicide ideation in 19.8% of girls in a school-based sample in Germany, and 10.8% had engaged in a suicide attempt. This was 9.3% and 4.9%, respectively, in males. The suicidal process may flow differently between boys and girls. Ideation peaks in mid adolescence in girls, whereas in boys a slow increase through late adolescence is seen [12]. Voss and colleagues [13] studied the onset of suicide attempts in a youth sample aged 21 and below. They reported an earlier onset of nonfatal attempts among boys but found that the cumulative incidence of nonfatal attempts was higher in girls from the age of 14 on.

While gender differences in the prevalence rates of suicide ideation, suicidal behaviour, and suicide death have been clearly established, the current literature offers limited insight into gender differences in the communication of suicidal thoughts throughout the lives of young suicide victims. This is a missed opportunity as enhancing our understanding of gender patterns in suicide-related communication can improve our recognition of the needs of suicidal boys and girls, which in turn is valuable for optimizing suicide prevention efforts. Recent findings from the reviews of Morken and colleagues [14] and Cha and colleagues [4] indicate that early intervention, such as school-based interventions, may prevent suicidal ideation and attempts in the short-term, and suicide attempts in the long term in youths and adolescents. The recognition of the youths' needs through their communication can guide strategies for early intervention. Therefore, this paper has suicide-related communication in relation to gender as its focal point.

In this article, we define personal suicide-related communication (SRC) as "the act of conveying one's own suicide ideation, intent, or behaviours to another person" [15], in which we include both online and offline communication. SRC carries a unique signalling function. Crucially, it reflects a capacity to ask for help [16, 17]. Evidently, SRC is not always a form of

help-seeking behaviour. For example, Frey and colleagues [15] suggest that a suicide-related notification aims to address practical considerations, such as planning a date or acquiring means, and suggests no clear intention to seek help. Frey also suggests that it is important to differentiate among different forms of SRC, such as disclosure, in which a subject intentionally communicates current or past suicidal thoughts and behaviour with the goal of seeking help or sharing their own personal history. Whether aimed at help-seeking or not, SRC appears to consistently indicate a desire to change a status quo unfavourable to one's wellbeing.

There is a scarcity of research into SRC of youths. Talking constructively about suicide can avert the risk of suicide [18]. Obviously, this requires an in-depth understanding of the way young people talk about suicide in the context of their social environment. Previous research on SRC has predominantly addressed the proportion of communicators or frequency of communications in a sample, which does not fully appreciate its complexity [19]. Moreover, these studies have presented various proportions of communicators, i.e. individuals who engaged in SRC, in samples of deceased youths, ranging from 5 to 85 percent [20–26]. Pompili and colleagues [19] suggest that the disagreement in findings may be largely explained by former studies' definition of SRC and methodological quality.

According to Rhodes and colleagues [27], males may be less predisposed to help-seeking behaviours. Alternatively, suicide ideation in young males may not be signalled properly due to ineffective or limited SRC. Gender differences can be observed in the use of health services by young suicidal persons. A Quebecois study [28], where health records of almost 1250 adolescents aged under 25 who died by suicide were examined, showed that significantly more female decedents [82.5%] had utilized healthcare services in the year prior to death by suicide, compared to male decedents [74.9%]. Females who died by suicide were also significantly more likely to have a mental health diagnosis, regardless of the care setting.

Rytterström and colleagues [29] emphasize that to better estimate the risk of suicide in youths, we should explore "the emotions behind the façade" and try to understand the "internal experience of life". Here lies a clear direction for rigorous qualitative research in order to obtain a deeper understanding of the what (content), how (form, medium) when (debut, frequency, last perceived event) and to whom (recipient) of SRC of young suicide victims. In the current study we conduct an in-depth exploration of SRC in the lives of young suicide victims in the Netherlands. This study aims to elucidate gender differences by comparing patterns in SRC of young males and females who died by suicide.

## Materials and methods

### Study design

Proxy informants of youths aged under 20 who lived in the Netherlands and died by suicide in 2017 were interviewed in a psychological autopsy study in 2019 [30]. Parents were the principal informants. Additional interviews were conducted with peers, teachers, and health care professionals. Respondents were contacted via their general practitioner. We obtained the contact data of the general practitioner from coroners who operate regionally and keep records of all suicide cases in the Netherlands. Secondary informants were recruited through the parents. The sampling strategy is further detailed by Mérelle and colleagues [30]. Interviews lasted approximately 2–3 hours and were conducted by an interviewer and a researcher (EB, ML). The interview team received a three-day training from a psychologist and an actor, aimed to get acquainted with the instrument and to master adaptive techniques to interview a vulnerable study population. A qualitative, explorative design was used to analyse interviews with next of kin (NOK). These were the parents and peers.

## Materials

Interviews were semi-structured. The instrument was largely based on instruments from international psychological autopsies in Belgium, Ireland, Norway, and the United Kingdom [20, 31–34] and consisted of two parts. NOK were invited in the first part to provide a narrative account of the youth's life, their relationship, and their own perceptions about key factors contributing to the suicide. The second part consisted of sections covering five pre-identified topics: [1] adolescence, [2] healthcare, [3] social media use and contagion effects, [4] sexual orientation and gender identity, and [5] religion and ethnicity. The aforementioned topics were based on an extensive body of evidence on risk factors for suicide and obtained by consensus from the research group and an advisory committee for the current study. Seven questions and corresponding follow-up questions were dedicated to SRC (e.g. *did the youth ever talk about dying, to you or anyone else*?). The interview was piloted in four test cases to assess its application in practice and make final changes. The final instrument, including the questions regarding SRC, has been added as a supplement.

## Analysis

For the qualitative analyses, interviews were recorded, transcribed verbatim and coded in several cycles using an ATLAS.ti (version 8.3) software package. In the first cycle, the main researcher (EB) coded the interviews. Initial coding was based on predetermined elements of SRC, as guided by the study's first research question: "*What are the debut, form, medium, frequency, content, recipient, and the last perceived events of suicide-related communications by youths who died by suicide*?" The variables as formulated in the research question were decided on *a priori* as relevant, operationalized into a coding sheet and integrated into ATLAS.TI. The second cycle was iterative. Additional codes were created, and existing codes were refined (e.g. "SRC medium: texting and messenger apps"). In the third cycle, a second researcher (PP) assessed the coded data, suggested changes, and marked conflicts. Lastly, proposed changes were presented to a third researcher (SM), and consensus was sought. The final coding sheet has been added as a supplement to this article and includes the current study's working definitions of SRC.

While implicit communications, both verbal and nonverbal, can be crucial signals, the extent to which they reflect suicidal intent can be uncertain using psychological autopsy data. Therefore, we have included only *explicit* communications throughout the course of life. The disclosure of suicidal feelings, or acquiring means to end one's life, for example, are an unequivocal reflection of suicide ideation. In contrast, only in the last months of the youths' life, we have also analysed *implicit* SRC, which were verbal communications and notable (changes in) behaviours that NOK perceived, in retrospect or at the time, as signals of suicide ideation or intent, or impending suicide. This was guided by the assumption that suicide ideation would have manifested itself by that time in a majority of cases.

We adapted the Constant Comparative Method [35] to examine patterns in SRC of young males and females and identify gender differences. We first explored individual SRC and thereafter compared differences and similarities in SRC within and between gender groups. To explore individual SRC, the first author created a matrix of communication events (i.e. interview data describing the occurrence of SRC). This comprised a brief account of the content, form, and recipient(s) of individual SRC events. Additionally, the occurrence of SRC events throughout the youth's life (e.g. debut, frequency, last recorded event) was determined. By detailing the collection of individual SRC events and creating an exhaustive account of the data, we aimed to reduce researcher bias [36]. Descriptive statistics were used to analyse the demographic characteristics of the sample and the number of recorded SRC events. Two-tailed t-tests were performed to examine gender differences in the occurrence of SRC events.

To subsequently compare SRC within gender groups, the first author amalgamated data into case-level data clusters. This entailed a summary of all recorded events of a case, which facilitated axial comparison. An example of a converged data cluster is illustrated below (Fig 1). Finally, we compared patterns in SRC between the male and female groups to answer the research question: *What gender differences can be observed in the suicide-related communications of youths who died by suicide in the Netherlands*?

### Ethical approval

The Medical Ethical Committee of Amsterdam UMC approved the study [registration number: 2018.651—NL68348.029.18]. All participants in this manuscript gave written informed consent.

## Results

Proxy informants of a sample of 35 deceased young people, 18 females and 17 males, were interviewed. We conducted 37 interviews with 54 parents and 18 interviews with 19 peers. The number of conducted interviews per case varied from one to four. We interviewed two NOK per case on average, which always included at least one parent.

Table 1 presents an overview of the SRC events as derived from analyses. SRC events were identified in 100 percent of youths in our sample. The mean number of SRC events reported per case throughout life was 22.80 (min. 3, max. 61), with a relatively high standard deviation (13.70). While the mean number of SRC events (explicit and implicit) was notably higher in female cases, this was not statistically significant. However, nearly twice as many explicit SRC events were recorded in girls, which was significant (p = 0.02).

The following sections address our findings from pattern analyses. Where applicable, the sections start by presenting general patterns identified in the SRC of youths. Thereafter, patterns in SRC of boys and girls will be specifically highlighted.

### Section 1: The occurrence and dispersion of SRC

The debut (i.e. the first recorded event) of SRC appeared to occur later in male cases than in female cases, usually between 14–16 years old, and was often more proximal to the suicide-act in male cases. A single exception was a boy who first said he wanted to die when he was five years old. In four out of 17 male cases, there were no recorded explicit SRC events (verbal, nonverbal, or written) until the last months before the suicide-act. In five other male cases, there were singular events in their lives prior to the last months. Common debuts of SRC in young males consisted of verbal SRC events. To a lesser extent we recorded debuts of nonverbal SRC in males, such as deliberate self-harm, nonfatal suicide attempts, or suicide information seeking. For several male cases, the debut of SRC concerned an implicit communication event in the last months alive.

Most female cases in the study sample, on the other hand, were characterized by various SRC events over a longer period preceding the suicide-act. This encompassed verbal

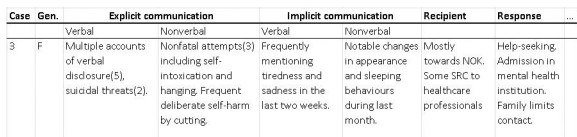

**Fig 1. Example case-level data cluster.**

**Table 1. SRC of sample cases.**

|  | All (n = 35) | Females (n = 18) | Males (n = 17) |
|---|---|---|---|
|  | Mean (SD) | Mean (SD) | Mean (SD) |
| Age | 17.0 (1.5) | 17.3 (1.3) | 16.7 (1.7) |
| Interviews | 1.5 | 1.5 | 1.6 |
| Respondents | 2.1 | 1.9 | 2.3 |
| SRC events | 22.8 (13.7) | 26.1 (15.5) | 19.3 (10.9) |
| Verbal SRC events | 6.6 | 6.9 | 6.2 |
| Nonverbal SRC events | 12.5 | 14.9 | 10.0 |
| Written SRC events | 3.7 | 4.3 | 3.0 |
| Explicit SRC events | 13.7 (10.3) | 17.6 (11.6) | 9.6 (6.8) |

expressions of suicide ideation or intent, as well as long-term deliberate self-harm, former nonfatal suicide attempts, suicide-related online behaviours, and written SRC. Half of the girls had a debut of SRC around the transition to high school, at 11–13 years old. The most common debut of SRC in girls was deliberate self-harm, followed by disclosure and nonfatal suicide attempts.

**Verbal SRC.** Verbal SRC events in boys rarely included straightforward disclosure of suicidal thoughts or clearly recognizable suicidal behaviour. Instead, boys frequently asked questions about death or reincarnation, spoke about death or suicide in a general sense, voiced their opinion about suicide, vaguely hinted at not being alive anymore (*e.g. "I am never going to be old anyway"*), or made jokes involving death or suicide that seemed random at the time. When the parents of one boy said they would go shopping and asked him if he needed anything, he replied: "*all I need is a rope*", which at the time his parents believed to be a joke. Only six of the 17 boys unequivocally spoke about suicide ideation or suicide plans in the years preceding their death.

Girls addressed suicidal feelings more directly in verbal SRC and made few cynical remarks or macabre jokes. Fifteen out of 18 girls had unambiguously disclosed their own suicide ideation or suicide plans to at least one other person. Notably, three girls did not verbally express their suicide ideation or suicide plans, of whom one engaged in deliberate self-harm and one had made a nonfatal suicide attempt. Eleven girls had had multiple conversations with their NOK about wanting to take their own life. Notably, these were mostly girls with complex psychopathology, such as clinical depression with comorbid eating disorder, who had been admitted to a mental health institution at least once in their life.

**Nonverbal SRC.** Deliberate self-harm was described in five out of 17 male cases. Two boys cut themselves, a method also reported in females, but there were also methods of deliberate self-harm unique in the male subjects, such as hitting objects with their hands until they bled or slamming their head against a wall. One boy deliberately jeopardized his physical integrity by ignoring treatment prescriptions for somatic symptoms, neglecting the concerns of NOK. Deliberate self-harm or nonfatal suicide attempts prior to the last months of their life were described in eight male cases, but many nonverbal communications were ambiguous, despite an inextricable link to death, dying, or suicide. One boy had carried around a miniature noose, which was found in his pocket when he died. His parents had reason to believe he had carried it for a long time and that he kept it around as "*a symbol [to remind or comfort him] of how he always could [end it]*". One boy went out with his bicycle in the middle of the night to visit the location where he took his own life one year later. When his parents confronted him with his behaviour at the time, he would not say why he went there.

In total, 13 out of 18 girls had engaged in some form of deliberate self-harm, including cutting, ingestion of objects, and self-starvation. The latter was exclusively found in girls. First events of deliberate self-harm occurred as early as 11 or 12 years old in our sample. Nearly all girls who initially self-harmed in early adolescence continued these behaviours until the moment of their death. In 11 girls, deliberate self-harm was followed by one or multiple nonfatal attempts. Ingesting an excessive amount of medication was frequently described by respondents as a first attempt. Over time, most girls proceeded to use more lethal methods for attempting suicide, such as auto-suffocation and hanging. This was clearly illustrated in one of the female cases.

*Parent*: *on December 12th, she made a first suicide attempt [. . .] she took 20 paracetamols. On February 15th, she attempted again, with 30 paracetamols. She started cutting more, and closer to her wrists. [. . .] In June, she made another attempt using 40 paracetamols.*

Respondents of a small number of female cases recalled other forms of explicit nonverbal SRC. One girl made a bucket list of things to do before her death, which she shared with a friend. Another wrote and sang multiple songs about dying with one of her friends and posted these on YouTube. Several girls, and one boy, developed an interest in emotional texts, images, songs, and/or videos relating to death or suicide. Girls selected online communities in which youths with mental health problems and suicide ideation aired their thoughts. Here they lurked at dark and depressing quotes and images (memes) from online contacts or created their own collection.

**Written SRC.** Written explicit communications about suicide or suicide ideation or intent were rare among males in the years preceding death by suicide. Particularly posting about mental health or suicidal thoughts in online communities was seldom mentioned. This was described in only two male cases. Three boys communicated suicide plans in WhatsApp messages to their mother or girlfriend. Notes of two boys detailing their struggle with suicidal thoughts were only found after their deaths.

By contrast, at least 13 of the girls described their suicide ideation or intent explicitly in diary entries, online posts, fictional stories involving dying or suicide, or in drawings related to suicide. According to respondents, most of the conversations in the online communities that the girls were part of were not about suicide. Nevertheless, multiple girls had reportedly written about suicide ideation to friends or strangers online and included pictures of their deliberate self-harm and *selfies* after nonfatal attempts. The latter behaviour was exclusively reported for girls who had been residents of a healthcare institution.

**SRC in the last months.** In our next step, we compared patterns of explicit SRC in the last months to those during life. Specific nonverbal SRC became more pronounced in the last months. Whereas six youths seemed to have prepared for their suicide in the year before suicide, this number had increased to 18 in the last months. These 18 youths enquired about methods on forums or social media, gathered information through search engines, scouted locations for a suicide, looked up train schedules, or acquired means for suicide, such as a rope or helium tanks. One girl set up her social media to keep posting after her death. A boy started a bucket list, like the aforementioned girl, but briefly before death. Crucially, preparations for suicide were discovered after death in eight of the 18 cases. In three cases it was unclear from the interview when preparations were discovered by NOK. Suicide notes were written by six boys and seven girls. Five other youths wrote a farewell message through a computer or in a text message. The other youths had not left a farewell message according to NOK.

Even proximal to the suicide-act, explicit SRC remained scarce in most boys. In two male cases with no history of SRC, there was a sudden onset and rapid escalation of both verbal and

nonverbal events in the last months, which happened parallel to sudden development of psychotic symptoms. One of these boys, who had never disclosed any suicidal feelings or reported suicidal behaviours, rapidly proceeded from communicating distress, stating: *"my thinking is driving me crazy"*, to openly discussing suicide methods with his mother, saying: *"I want to talk to you tonight. I want to tell you how I'll do it."*

**Implicit communications.** Respondents elucidated various implicit communication events in the last months which they associated with the suicide. Most youths verbalized their emotional distress and sadness, were increasingly tired, changed their sleeping patterns, or isolated themselves from their loved ones. Furthermore, multiple youths deleted social media accounts or exited WhatsApp groups. Respondents of a small number of cases recalled notable changes in appearance, academic performance, and religious behaviours in the last months. A distinctive phenomenon in our interview data was a perceived upswing of youths. Varying from one day to several weeks before death, 13 youths (seven boys, six girls) seemed more "relaxed" or "happy" to our respondents than before. Some youths were said to have invested more time in social contacts in the period just before their death. This perceived upswing was most common in the days briefly before the suicide-act.

> *Parent*: *"That Sunday, I came back from the theatre. She was sitting on the couch; she had colour in her cheeks. She hadn't looked that good in years. She was lively, had taken a shower, everything was shaved, she had washed her hair. [. . .] we had diner together, it was a very relaxing day."*

Gender differences in the occurrence of implicit communications were limited. In hindsight, it became clear to NOK that six girls had been actively, yet implicitly, saying goodbye. They repeatedly spoke out their love and affection for NOK, ordered excessive numbers of gifts for NOK, met up with friends they had not seen for a long time, or hosted longer visits of their NOK in the clinic. One girl had rearranged her entire bedroom to represent memories from her life, using pictures and other objects. These behaviours were not recorded in boys.

## Section 2: Content of communications

Recorded verbal and written SRC events provided insight into the youths' own vocabularies. In 25 out of 35 cases, NOK described the content of verbal and written SRC events, by paraphrasing or quoting the youth. Central themes in the content of explicit SRC throughout life were universal for boys and girls. Explicit SRC events reflected the youths' distress, depressive feelings, and hopelessness and revealed how suicide ideation started and became more pressing over time. The young people additionally spoke out their fears and, crucially, many asked for help and support.

Young boys were less inclined than girls to disclose suicide ideation or ask for help in the content of SRC during life. They seldom expressed a need for professional healthcare. Several boys who had received care described negative feelings or an opposing attitude towards care. A parent illustrates: *"He [son] said: so, I'm supposed to go here and just do a bit of clay modelling with more of those people, and that's supposed to make me feel better? He [son] said: forget it. I am not going."* Another mother argued that sitting face-to-face with a young psychologist to talk was not effective for her son, although it did not keep him from going: *"He would mope every time, but he went every week. He was always compliant. Although he did walk out a few times. [. . .] They should have connected better with him [. . .] but these were not the right people for that, these were young female therapists; he needed something else."* NOK's own attempts to talk to boys about emotional wellbeing were strained when trying to get to the bottom of their

distress. As one mother illustrated: *"he always said: 'I am really losing my mind.' Then I would sit next to him, ask: what does that mean? [. . .] He would not say anything. [. . .] When I tried to delve deeper, he said: 'Stop it! I don't want to talk about it anymore.'"* Something another boy wrote in a note found after he died suggested an avoidant coping strategy: *"nobody notices, not even my dear mother"*.

By contrast, the content of explicit SRC events in girls in our sample seemed more purposeful compared to males in that they more often clearly addressed suicide ideation or intent, expressed their will to seek professional help, or focused on coping or the difficulty thereof. Several girls had asked for support from friends and family, or actively addressed unmet health needs.

> *Parent*: *"She was 15 and had just started treatment. It didn't work out, and she frequently entered emergency services. She opened up about it, too. She said that she wasn't as strong as me and that she couldn't do it alone."*

**Content of SRC in the last months.** In the content of SRC events in the last months, there was increasing emphasis on emotional detachment from friends and family, hopelessness, negative perceptions about the future, and readiness to die. Several youths stated that they were only alive for others.

> *Friend*: *"And then she told me: 'I really cannot go on living for you.' That got to me. I asked her: 'Can't we try to do anything fun together, or make it work, or try something that will make it better?'. 'No, no' [she said]"*

The topics of verbal communications shifted. Eight youths started discussing suicide methods. Some shared intimate details about previous nonfatal attempts, asked for permission to die, or made threats to take their own life, which put a great strain on NOK. The topic of euthanasia or assisted suicide, which had formerly been reported in three girls, was addressed by four more youths (two males, two females) in the last months. Youths would discuss how they valued dying in a "proper" or "clean" way. Lastly, it stood out that five girls and two boys made various claims in the last months which seemed to suggest that they believed suicide was not a definite ending to their existence.

> *Parent*: *"If you believe that dying is the end of all things, that everything stops and that there is nothing, then the step to take your own life is much more frightening. That was not at all how she saw it. She felt like: 'I'm going to my grandfather [in the afterlife]'."*

Notably, negative feelings about healthcare emerged in SRC of girls in the last months. Whereas several boys seemed opposed to reaching out to professional help throughout their lives, a group of five girls who had received various forms of therapy stated in the last months that they had lost faith in the effect of therapy. As a result, they believed themselves to be "beyond help". Parents of a girl illustrate: *"We were invited to the therapist's office, who again focused on healing or at least dealing with the problems and looking forward. But she [daughter] formally stated in that conversation that she wanted to cease living. [. . .] that she wanted to stop treatment."*

The content of suicide notes or other farewell messages were provided by NOK of 14 youths. Youths mostly expressed their love for NOK and tried to minimize potential feelings of guilt about the suicide. As a parent recalled: *"[he wrote] that he had a good life. That he was*

*happy. That this was the only option for him. But with a message to everybody to not grieve, to be happy."* Youths further described how they could no longer deal with the pain, provided justification for their death, or made practical arrangements. In contrast with the content of preceding verbal and written SRC, we identified no notable gender differences in the content of suicide notes.

**Content of implicit communications.** The content of implicit communications, where suicide was not directly mentioned, was diverse. Most youths, regardless of gender, highlighted emotional distress, predominantly sadness, anxiety, and insecurity. According to NOK, however, boys more frequently ventilated their frustrations and felt that they did not belong or that they were "stuck". A few respondents recalled increased verbal aggression in male youths. In contrast, NOK mentioned that girls verbalized perpetual guilt and thought of themselves as a burden. As a mother illustrated: *"She [daughter] did not know what the matter was. She constantly felt guilty. That feeling of guilt and those crazy jumps [in behaviour] and saying things that she did not understand why she said. [. . .] She [daughter] said: mostly that feeling of guilt towards everyone".*

## Section 3: Recipients of SRC events

Recipients can play a crucial role in early signalling and suicide prevention. In a group of five boys, parents were the only recipients of SRC events, and in six other cases it was only the parents and a close friend or a romantic partner. In contrast, there were more varied recipients of SRC events for girls. These included parents, friends, classmates, healthcare personnel, peers with similar problems online, and fellow residents of mental care institutions. The latter two were unique recipients of female SRC events.

NOK's interpretation of SRC events and NOK's response to SRC appeared to parallel the identified gender patterns in SRC. Additionally, these themes reflected the challenges in talking about suicide with young boys and girls from the perspective of NOK.

**NOK's interpretation of SRC events.** SRC was predominantly interpreted by NOK as a signal of distress or a cry for help and prompted a response such as emotional support or seeking professional help together. Both explicit and implicit communications affected NOK's perceptions about suicide risk. In several cases, the seemingly overwhelming number of SRC events led to a fatalistic view among NOK, which one parent summarized thus: *"It wasn't a question of 'if' but 'when' it would happen."* In other cases, NOK explained how they normalized SRC events because they were afraid to acknowledge the reality of suicide. As a mother illustrated: *"There is no way you can live with the idea that she'd do it because that would drive you insane. So, you just have to live by: 'I am doing the best I can', and 'she won't do it'."* Several youths denied suicide ideation and/or suicide plans when asked by NOK, which lowered NOK's perceived risk of suicide.

Some gender differences were observed across interpretations of SRC events by NOK. Due to their isolated occurrence and sometimes vague or macabrely humorous presentation, NOK found the seriousness of SRC events of young males difficult to determine. This difficulty was not mentioned by NOK of females. Furthermore, NOK of nine boys explained that while they did notice that the youth seemed to struggle, appeared unhappy, or behaved differently than before, they never expected suicide to be a realistic outcome of these dynamics. Crucially, in the absence of unequivocal signals of suicide ideation or intent in boys, NOK perceived no mental health needs to respond to at the time.

> *Parent: "In hindsight I sometimes think maybe his 'not feeling happy' was stronger than we noticed at the time. But you just don't know . . . Only when looking back. At that moment there was no reason to worry."*

Sometimes SRC events, in particular deliberate self-harm and nonfatal suicide attempts, were perceived by NOK as attention seeking, a dramatic display of emotions, or a way to achieve a desired outcome. *Brother*: *"Many of her attempts had been half-hearted. [. . .] Sometimes it seemed she was doing it mostly for the attention."* This was observed in SRC events of at least seven girls who showed frequent suicidal behaviour. It must be noted that NOK of these girls generally still sought help despite their own beliefs.

**NOK's response to SRC events.** In response to most SRC events of both boys and girls, NOK tried to engage in open conversation, recommended the youth to seek professional care, or sought help together. However, parents and peers mentioned that it was challenging to talk about suicide with the young person. They attributed this to stigma, which was reflected in feeling uncomfortable or inadequate to address such a difficult subject, and some NOK were afraid to aggravate or normalize the problems if they were to discuss them with the young person.

> *Parent*: *"How are you supposed to react [to the expression of suicidal feelings]? It scared the hell out of me to hear him say things like that. I said nothing. You become immobilized."*

> *Parent*: *"Whenever I communicated about it [suicide], it was through WhatsApp. [. . .] I asked her some things through WhatsApp that I did not dare ask face-to-face."*

While considering that support and help-seeking were most common, the interview data did suggest that NOK of boys would more commonly respond with "tough love". In seven male cases of whom multiple SRC events were noted, NOK did not respond, hardly asked questions, avoided the conversation, stressed the youth's own responsibility, or (in hindsight) downplayed the seriousness of SRC. One parent told us: *"I said: [NAME], you called this out on yourself, you alone are the reason you are here [in a psychiatric hospital]. [. . .] Now you've done it. Because they are going to lock you up now, is that what you want?"* Another parent remembered: *"that conversation we had at the dinner table; he was crying. He had frequent headaches. [. . .] later it turned out he could not sleep for nights, or only a few hours, because of those headaches. I remember that we used to say: 'keep your head up, take an aspirin.' Looking back, we would never have done that, of course."*

In three male cases, NOK responded with physical aggression. When one of the boys hit his head against a wall, his father would pick him up and shake him until he stopped. One brother described how he tried to help his depressed, suicidal brother in various ways but ended up physically assaulting him when words failed.

> *Brother*: *"He was in his bedroom, and I beat him up quite badly, just because I felt powerless. I tried to help; I tried all kinds of things. One time, a colleague and I made a whole list of positive things about him [victim] because he thought he wasn't good enough. [. . .] you feel so powerless and there's nothing you can do, so I barged into his room and we had a bad fight."*

Instead, SRC events of girls more often triggered conversations. NOK of 14 girls had asked follow-up questions to elucidate details, and seeking help together was common. Notably, data suggested that deliberate self-harm and nonfatal suicide attempts were strong cues for NOK to discuss suicide ideation or seek professional help. Adversely, frequent occurrence of deliberate self-harm and attempts had NOK constantly worry, which could evoke negative feelings about their relationship with the youth. One mother well remembers her struggle: *"has she been on WhatsApp? Oh, she's been offline for 45 minutes. . . Who has seen her since? Has someone seen her on Instagram? You know. . . that's no way to live. It's impossible."* Consequently, some NOK restrained contact.

*Peer*: *"[I received] a weekly phone call, in which she said: 'I tried to take my life again'. I thought to myself: I have a life too. I am in school, I go to work, I cannot handle all this. We grew apart. [. . .] Honestly, I was not looking forward to visiting her [in a psychiatric hospital]. You sit in front of each other . . . What do you say? My life is moving forward, yours is at a halt. That felt awkward."*

## Discussion

This study aimed to explore patterns in SRC of youths and to examine gender differences in these patterns. Almost 800 SRC events of 35 youths who died by suicide in the Netherlands in 2017 were explored by means of qualitative analyses.

### Frequency and debut

We have identified SRC events in all youths in our sample. Results show that females had an earlier debut of SRC and significantly more *explicit* SRC events. Most debuts of SRC in girls occurred at the start of puberty, whereas the first SRC events of boys more frequently occurred in the last months of their lives. Marttunen and colleagues [26] found that adolescent males without a diagnosis for mental health problems communicated suicidal thoughts shortly before suicide. Aligning this, fewer boys than girls in our sample had diagnosed mental health problems [37]. Finally, our findings concerning SRC debut show parallels with the onset of suicide ideation in boys versus girls [4, 12], which may suggest that boys and girls engage in SRC briefly after first ideation.

There may be several reasons for the gender discrepancy in the frequency of explicit SRC events. Logically, the fact that the SRC debut in boys was often proximal to suicide narrows the timeframe in which SRC events could have occurred. Second, the difference in the frequency of nonverbal events may be explained by the propensity of girls to engage in deliberate self-harm and nonfatal suicide attempts more than boys [4], which could also be observed in our sample. De Beurs [38] suggests that the relatively higher prevalence of psychopathology such as depression and ideation in young girls, as well as trauma from sexual abuse at a young age, may contribute to this propensity. Crucially, there is limited evidence showing which prevention and treatment strategies work for youths with chronic suicidal behaviours [18]. Lastly, respondents explained that boys refrained from conversations about emotional distress overall and believed this probably has hampered verbal SRC. Husky and colleagues [39] report that being male is indeed associated with non-disclosure of suicide ideation, and De Luca and Wyman [40] confirmed this in a study among Latino adolescents. Nolen-Hoeksema and Girgus [41] propose that the emotion-focused coping of girls may be more focused on communication and rumination, whereas the coping of boys may be more distraction-oriented. Corroborating this, girls are more likely to disclose emotions to parents and friends than boys [42]. Alternatively, male beliefs about masculine norms may influence non-disclosure of suicide ideation and affect help-seeking in males [38, 43], but it is unclear how this may affect youths.

### The last months prior to suicide

We identified notable changes in explicit SRC during the last months, irrespective of gender. The emphasis of communications shifted towards practical considerations of suicide, such as discussing the methods or talking about a planned date for the suicide. NOK furthermore associated a plethora of implicit SRC events in the last months of the young people's lives with the suicide. What stood out specifically was the perceived upswing of 13 youths briefly before

suicide, which was universal for boys and girls. Our study cannot fully explain the underlying mechanisms of this upswing but provides evidence of its existence. Parents in our sample suggested that the upswing might indicate that their child had made a final decision. The phenomenon has been proposed by mental healthcare professionals as a signal of impending suicide in youths [18]. There is, however, a lack of conclusive evidence in the literature and no firm conceptual basis. Antecedents to youth suicide as observed in SRC can be subtle but may provide implications for signalling suicide ideation and arguably represent a closing window of opportunity for intervention. Academic consensus on an operational definition would facilitate research of antecedents such as the perceived mood upswing.

## Content of SRC

The content of explicit SRC events in boys was often ambiguous. Additionally, several events in young males reflected an opposing attitude towards care. Contrastingly, the content of SRC events of girls seemed more purposeful, showing emphasis on coping and help-seeking, with a negative attitude towards care developing only in the last months, after trying extensive therapy. This corroborates cross-national research which shows that young males suffering suicide ideation are less predisposed to seek professional mental health for their mental distress [44, 45]. It is possible that boys' use of more macabre jokes and vague terminology regarding suicide ideation reflect negative perceptions about help-seeking. These in turn may result from self-imposed gender-role restrictions among males, in which help-seeking is a sign of weakness and at odds with masculinity. This relates back to previous research into disclosure of depression by adult males [27, 46, 47]. In our study, vague communications seemed less likely to result in support or proactive help-seeking by NOK. This stresses the importance of encouraging young males to be unequivocal in their expression of suicide ideation, which may go hand in hand with help for NOK and professionals in learning how to ask boys more upfront questions about their wellbeing, particularly when boys seem ambiguous about their mental health.

## Recipients of SRC

Girls in our sample directed SRC towards relatively more varied recipients than boys. They spoke to friends and companions with similar feelings, both online and offline. Additionally, as more girls were admitted into a mental health institution, healthcare professionals were common recipients of SRC events of girls. Boys who verbalized their suicidal thoughts generally limited any communications to parents, romantic partners, and a few close friends. There is evidence that young people are more likely to express suicide ideation or intent to peers than to adults [6, 20, 48]. Since we are using the psychological autopsy method with parents as primary proxy informants, we cannot confirm this. However, peer interviews provided additional insights into communication events in specific settings, such as in mental health institutions and online, which is also concluded by research of Looijmans and colleagues [49].

Additional insights emerged concerning NOK's interpretation of SRC and their response to specific events. NOK acknowledged their inability to assess the seriousness of SRC events for various reasons, such as the ambiguity of SRC events in males and the overwhelmingness of SRC events in girls. Both appeared to lead to desensitization for future SRC events. Furthermore, it was difficult for NOK to differentiate between behavioural changes due to psychosocial development in adolescence and potential nonverbal SRC events. NOK's responses to SRC varied. Open conversation, support, and help-seeking were the most common. In the male cases, we found that NOK more often did not respond to SRC events, emphasized the youths' own responsibility, or responded with physical aggression.

### Strengths and limitations

Interviews with NOK provided a rich, detailed account of the SRC events of the deceased youths. The sample had a demographic profile that was similar to the total group of 81 youths who died by suicide in 2017 in the Netherlands [50, 51]. Furthermore, the male to female ratio was near 1:1, which, although not reflecting actual figures of suicide [50], was suitable for an in-depth exploration of SRC and to highlight gender differences.

However, some limitations of this study must be considered. First, this psychological autopsy was conducted without a control group due to the nature of the original research. Therefore, we do not know how the identified patterns of SRC in our sample would relate to patterns of SRC found in young suicide survivors. Second, the small sample size limits the generalizability of conclusions based on statistical comparisons. Third, parents were asked to recruit secondary informants, which could have led to selection bias of NOK with a shared narrative. Lastly, interviewing proxy informants 1.5–2.5 years after the suicide may result in recall bias as well as socially desirable responses concerning SRC events. More specifically, NOK may have forgotten SRC events or feel conflicted to talk about specific events that they think might reflect negatively on them. Pouliot and De Leo [52] suggest that *"Emotion-related response bias may pose as well a challenge to the reliability of life events assessment based on proxies' report"* in psychological autopsy studies, which is likely to affect recollection of SRC events in a similar manner. However, Assink [53] postulates that negative life events are more consistently recollected by people compared to neutral or positive events, which would include NOK's memories about precipitating factors and antecedents for the suicide of their loved ones. Including multiple respondents can furthermore reduce such forms of bias, but secondary informants were not interviewed in all cases. Perhaps consequentially, we have obtained limited insight into online SRC events, particularly in boys. Several boys in our sample used videogame chat servers, such as Discord or Steam Chat, about which parents had little knowledge. Dutch research suggests that parents are not fully aware of their children's online risk behaviours, such as cyber-bullying [54]. Furthermore, children are considered *digital natives* and may be well equipped to erase tracks of their online SRC.

### Conclusions

To the best of our knowledge, this is the first study to conduct a qualitative analysis of suicide-related communication events of young people who died by suicide and to explore gender differences using psychological autopsy data. Gender-specific patterns in suicide-related communications were identified. The early debut of suicide-related communication in girls in our sample highlights the importance of early screening and prevention efforts in young girls. Ideally, specific attention must be drawn to establishing their needs earlier in the suicidal process to facilitate improved personalized care. The late debut and ambiguous presentation of suicide-related communication in boys may guide NOK and professionals to adapt communication strategies, for example by obtaining unequivocal confirmation or disconfirmation of suicide ideation. The overall findings underline the importance for NOK and professionals who work with potentially suicidal youths to be aware of gender differences in suicide-related communication and coherent challenges in communication. The potential causality between the frequency, form, or content of suicide-related communication and a young person's likelihood of receiving care warrants further inquiry. Lastly, we encourage future research to expand on the topic of online suicide-related communication and the potential for suicide prevention on digital platforms.

## Supporting information

**S1 File. Interview instrument parents.**
(DOCX)

**S2 File. Code list and coding approach.**
(DOCX)

**S3 File. COREQ checklist.**
(DOCX)

## Acknowledgments

We would like to express our gratitude to all next of kin who have participated in this study. They have suffered a great tragedy and were willing to share their intimate, insightful stories with us. We believe these stories are of great value in informing suicide prevention strategies.

## Author Contributions

**Conceptualization:** Elias Balt, Saskia Mérelle, Diana van Bergen, Renske Gilissen.

**Data curation:** Elias Balt, Milou Looijmans.

**Formal analysis:** Elias Balt.

**Funding acquisition:** Renske Gilissen.

**Methodology:** Elias Balt, Saskia Mérelle, Diana van Bergen.

**Project administration:** Elias Balt.

**Supervision:** Arne Popma.

**Validation:** Saskia Mérelle, Pommeline van der Post.

**Writing – original draft:** Elias Balt.

**Writing – review & editing:** Saskia Mérelle, Diana van Bergen, Renske Gilissen, Pommeline van der Post, Milou Looijmans, Daan Creemers, Sanne Rasing, Wico Mulder, Lieke van Domburgh, Arne Popma.

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
