## [Decision Letter · Decision Letter 0]

1 Mar 2021

PONE-D-20-36446

Gender differences in suicide-related communication of young suicide victims

PLOS ONE

Dear Dr. Balt,

Thank you for submitting your manuscript to PLOS ONE. After careful consideration, we feel that it has merit but does not fully meet PLOS ONE’s publication criteria as it currently stands. Therefore, we invite you to submit a revised version of the manuscript that addresses the points raised during the review process.

Please be advised that submitting a review does not guarantee acceptance.

We look forward to receiving your revised manuscript.

Kind regards,

Vincenzo De Luca

Academic Editor

PLOS ONE

Journal Requirements:

4. We note you have included a table to which you do not refer in the text of your manuscript. Please ensure that you refer to Table 1 in your text; if accepted, production will need this reference to link the reader to the Table.

Reviewers' comments:

Reviewer's Responses to Questions

**Comments to the Author**

1. Is the manuscript technically sound, and do the data support the conclusions?

Reviewer #1: Yes

2. Has the statistical analysis been performed appropriately and rigorously? 

Reviewer #1: Yes

3. Have the authors made all data underlying the findings in their manuscript fully available?

Reviewer #1: Yes

4. Is the manuscript presented in an intelligible fashion and written in standard English?

Reviewer #1: Yes

5. Review Comments to the Author

Reviewer #1: General comments:

This is a well written paper that addresses an important topic, re: prevention strategies that can stem the rising trend of youth suicide worldwide. Specifically, it is an exploratory study of gender differences in adolescents communications about suicide leading up to their suicide. Research on suicide related communication is area of early intervention about which little is known currently. Evidence from a recent systematic review show that early intervention can prevent suicidal ideation and attempts in the short term and suicide attempts in the long term (Morken et al, 2020). This is the larger background that this current study has to be situated. A major shortcoming of the paper is the authors decision to not report interview data of the 4 "non-native" suicides in their study, as immigration status and ethnicity are often risk factors for youth suicide, and given the fact that the Netherlands has a sizeable immigrant population (10%).

Introduction: The research problem, study rationale, and study aims are clearly articulated. The background literature reviewed is relevant, and includes recent work on suicide related communication in adolescents.

Methods: The psychological autopsy methodology is well established in completed suicides research to investigate putative risk factors. Given the study aims, use of qualitative explorative semi-structured interviews is appropriate, as is use of convenience sampling as documented in an earlier report (Mérelle et al, 2017). It would be helpful to readers if the authors include a brief summary of their sampling strategy and mode of recruitment of study participants. Data analysis methods (thematic content analysis, constant comparative approach) are appropriate and and adequately described.

Results: results are clearly presented

Discussion: the discussion is relevant to the study findings, the authors suggest future research directions, and they adequately address study limitations

References

1. Morken, I. S., Dahlgren, A., Lunde, I., & Toven, S. (2019). The effects of interventions preventing self-harm and suicide in children and adolescents: an overview of systematic reviews. F1000Research, 8.

2. Lipsicas, C. B., & Mäkinen, I. H. (2010). Immigration and suicidality in the young. The Canadian Journal of Psychiatry, 55(5), 274-281.

3. Mérelle, S., Van Bergen, D., Looijmans, M., Balt, E., Rasing, S., van Domburgh, L., ... & Popma, A. (2020). A multi-method psychological autopsy study on youth suicides in the Netherlands in 2017: Feasibility, main outcomes, and recommendations. PLoS one, 15(8), e0238031.

6. PLOS authors have the option to publish the peer review history of their article (what does this mean?). If published, this will include your full peer review and any attached files.

Reviewer #1: **Yes: **Ademola Adeponle

---

## [Author Response · Author response to Decision Letter 0]

2 Apr 2021

Authors’ response to the academic editor’s comments

Please ensure that your manuscript meets PLOS ONE's style requirements, including those for file naming. The PLOS ONE style templates can be found at: 

Reply: We have reviewed the manuscript to ensure it meets the PLOS ONE style requirements. Firstly, we have ensured that the files for our resubmission have been named in accordance with the indicated requirements of PLOS ONE. 

- ‘Manuscript’

- ‘Revision with Track Changes’

- ‘Response to reviewers’

Secondly, changes to the style in the manuscript body text have been marked accordingly and can be viewed in the document titled ‘Revision with Track Changes’. We hope that these changes concur with the academic editor’s intended sections.

- Added reference to table 1 in manuscript body (page 8, line 190)

- Replaced dash with dot in the Table 1 title (page 8, line 195)

- Changed font of table and figure titles to normal body text instead of header level 2 (page 8, line 180, line 195)

- The PACE tool was used to ensure that the image file meets PLOS ONE image requirements

1. We note that you have indicated that data from this study are available upon request. PLOS only allows data to be available upon request if there are legal or ethical restrictions on sharing data publicly. For more information on unacceptable data access restrictions, please see http://journals.plos.org/plosone/s/data-availability#loc-unacceptable-data-access-restrictions. 

a. If there are ethical or legal restrictions on sharing a de-identified data set, please explain them in detail (e.g., data contain potentially sensitive information, data are owned by a third-party organization, etc.) and who has imposed them (e.g., an ethics committee). Please also provide contact information for a data access committee, ethics committee, or other institutional body to which data requests may be sent.

b. If there are no restrictions, please upload the minimal anonymized data set necessary to replicate your study findings as either Supporting Information files or to a stable, public repository and provide us with the relevant URLs, DOIs, or accession numbers. For a list of acceptable repositories, please see http://journals.plos.org/plosone/s/data-availability#loc-recommended-repositories.

Reply: In the Editorial Manager, we have indicated that the data are restricted.

a. We would like to specifically note that the interview data cannot be shared publicly because of ethical restrictions: the dataset contains potentially identifying and sensitive information. The Medical Research Ethics Committee (MREC) of Amsterdam UMC has imposed this restriction (registration number: 2018.651 – NL68348.029.18). For more information, the committee may be contacted through: metc@vumc.nl.

b. As the data is restricted, this comment does not apply.

2. Please amend your list of authors on the manuscript to ensure that each author is linked to an affiliation. Authors’ affiliations should reflect the institution where the work was done (if authors moved subsequently, you can also list the new affiliation stating “current affiliation: ____” as necessary).

Reply: The co-authors have been asked to review their affiliations. The list of authors has been amended, to ensure that each author has been linked to an affiliation which reflects the institution where the work was done. We noted that one affiliation was missing for author Pommeline van der Post. This err has been corrected in the manuscript (page 1, line 5). We corrected the affiliation of two other researchers (page 1, line 8-9 and line 11).

3. We note you have included a table to which you do not refer in the text of your manuscript. Please ensure that you refer to Table 1 in your text; if accepted, production will need this reference to link the reader to the Table.

Reply: We thank the academic editor for notifying us that we did not refer to Table 1 in the manuscript. We have included a reference to Table 1 in the manuscript body text (page 8, line 190).

Authors’ response to the reviewer’s comments

Reviewer #1

“This is a well written paper that addresses an important topic, re: prevention strategies that can stem the rising trend of youth suicide worldwide. Specifically, it is an exploratory study of gender differences in adolescents’ communications about suicide leading up to their suicide. Research on suicide related communication is area of early intervention about which little is known currently.”

Response: We thank the reviewer for complimenting our work. 

“Evidence from a recent systematic review shows that early intervention can prevent suicidal ideation and attempts in the short term and suicide attempts in the long term (Morken et al, 2020). This is the larger background that this current study has to be situated.” 

Response: We thank the reviewer for this important contribution from recent literature. We find that these recent findings provide a significant addition and help us to better delineate the context, relevance and aims of our research. We therefore included a reference to the work of Morken and colleagues in the introduction of our revised manuscript (page 4, lines 74-78):

‘Recent findings from reviews of Morken and colleagues [2020] and Cha and colleagues [2018] indicate that early intervention, such as school-based interventions, may prevent suicidal ideation and attempts in the short term, and suicide attempts in the long term in youths and adolescents. The recognition of youths’ needs as reflected in their communication can guide strategies for early intervention.’ 

“A major shortcoming of the paper is the authors decision to not report interview data of the 4 "non-native" suicides in their study, as immigration status and ethnicity are often risk factors for youth suicide and given the fact that the Netherlands has a sizeable immigrant population (10%).”

Response: We agree with the reviewer that it is crucial to better understand the suicidal process and coherent risk factors in youths with a migration background. In total, 25 of the 81 youths (31%) who died by suicide in the Netherlands in 2017 had an immigration background (Gilissen et al, 2018). Indeed, by having been able to only include four youths with a migration background into the sample, we are limited in our understanding of the suicide-related communication of these youths, as well as in our overall findings concerning the role of cultural and migration factors in suicidality of these youths (Mérelle et al, 2020). 

The authors would, however, like to express that the four youths with a migration background have in fact been included in the analyses of the current research. Our findings are based on all 35 youths in the sample, including non-Dutch youths as well as Dutch native youths. Several fragments of interview data as presented in the manuscript refer to a youth with a migration background.

We included four cases with a migration background, which we defined as having at least one parent who was born outside of the Netherlands. They were three girls and one boy. Two youths had a Western migration background, and two youths had a non-Western migration background. Only one youth had two parents with a migration background. Notably, of the three youths with a Dutch and a non-Dutch parent, the Dutch parent participated in the interview, which may have limited the cultural sensitivity of the findings. The four youths had diverse life stories, and we found no evidence of cultural factors specifically related to the suicide-related communication of these youths. Deriu and colleagues (2018) state that intergenerational conflicts and the family environment, such as poor communication and lack of parental support, can be important risk factors for suicide ideation and attempts in adolescents with a migration background. Conceivably due to our limited inclusion of youths with a migration background, we found no evidence of such or similar problems reflected in the youths’ suicide-related communication. 

The main lesson we took from our research is how crucial it is to adapt recruitment strategies effectively to reach individuals bereaved by the suicide of a victim with a migration background in the Netherlands. Research from Germany (Reiss et al, 2014) shows benefits and drawbacks of a community-based approach to improve inclusion of participants with a migration background. The strategy leads to higher inclusion and a heterogeneous sample, but it requires more staff and is costly. Others (Fête et al, 2019) suggest that a communication strategy and a snowball sampling method was key to recruit participants with a migrant background. In future psychological autopsies, we will consider such strategies to improve recruitment of participants with a migration background.

“Introduction: The research problem, study rationale, and study aims are clearly articulated. The background literature reviewed is relevant and includes recent work on suicide related communication in adolescents.”

Response: We thank the reviewer for the positive assessment of the introduction section. 

“Methods: The psychological autopsy methodology is well established in completed suicides research to investigate putative risk factors. Given the study aims, use of qualitative explorative semi-structured interviews is appropriate, as is use of convenience sampling as documented in an earlier report (Mérelle et al, 2017). It would be helpful to readers if the authors include a brief summary of their sampling strategy and mode of recruitment of study participants. Data analysis methods (thematic content analysis, constant comparative approach) are appropriate and adequately described.”

Response: For conciseness, we opted to refer to the work of Mérelle and colleagues (2020) for the sampling procedures. However, we agree with the reviewer that including a summary of the sampling strategy and mode of recruitment would add to the comprehensiveness for the reader. As such, we have provided a summary of our approach in the revised manuscript (page 5-6, lines 120-123). The added section summarizes the following steps briefly:

- We contacted coroners of municipality health services in the Netherlands. They keep records of suicides 

- Coroners provided us with the contact details of general practitioners of whom a patient aged <20 had died by suicide in 2017

- We contacted the respective general practitioners, and asked them to inform the parents of the deceased youth about the research

- Parents who showed interest received a patient information folder

- Secondary informants (peers, relatives, teachers, and healthcare professionals) were recruited with the help of the parents

“Results: results are clearly presented”

Response: We thank the reviewer for the positive assessment of the results section. 

“Discussion: the discussion is relevant to the study findings, the authors suggest future research directions, and they adequately address study limitations.”

Response: We thank the reviewer for the positive assessment of the discussion section. 

References

Morken, I. S., Dahlgren, A., Lunde, I., & Toven, S. (2019). The effects of interventions preventing self-harm and suicide in children and adolescents: an overview of systematic reviews. F1000Research, 8.

Cha CB, Franz PJ, M. Guzmán E, Glenn CR, Kleiman EM, Nock MK. Annual Research Review: Suicide among youth – epidemiology, (potential) etiology, and treatment. J Child Psychol Psychiatry Allied Discip. 2018;59(4):460–82.

Gilissen R, Merelle S, Franx G, Popma A. Zelfdoding bij jeugd tot 20 jaar: eerste duiding van de cijfers. 2018. Ministry Report. 

Mérelle, S., Van Bergen, D., Looijmans, M., Balt, E., Rasing, S., van Domburgh, L., ... & Popma, A. (2020). A multi-method psychological autopsy study on youth suicides in the Netherlands in 2017: Feasibility, main outcomes, and recommendations. PLoS one, 15(8).

Deriu V, Benoit L, Moro MR, Lachal J. Idées suicidaires et tentatives de suicide à l’adolescence en contexte migratoire [Suicidal thoughts and suicide attempts in adolescence among migrants]. Soins Psychiatr. 2018 May-Jun;39(316):22-26.

Reiss K, Dragano N, Ellert U, Fricke J, Greiser KH, Keil T, Krist L, Moebus S, Pundt N, Schlaud M, Yesil-Jürgens R, Zeeb H, Zimmermann H, Razum O, Jöckel K, Becher H. Comparing sampling strategies to recruit migrants for an epidemiological study. Results from a German feasibility study, European Journal of Public Health, Volume 24, Issue 5, October 2014, Pages 721–726. 

Fête, M., Aho, J., Benoit, M. et al. Barriers and recruitment strategies for precarious status migrants in Montreal, Canada. BMC Med Res Methodol 19, 41 (2019). https://doi.org/10.1186/s12874-019-0683-2.

---

## [Decision Letter · Decision Letter 1]

10 May 2021

Gender differences in suicide-related communication of young suicide victims

PONE-D-20-36446R1

Dear Dr. Balt,

We’re pleased to inform you that your manuscript has been judged scientifically suitable for publication and will be formally accepted for publication once it meets all outstanding technical requirements.

Kind regards,

Vincenzo De Luca

Academic Editor

PLOS ONE

Additional Editor Comments (optional):

Reviewers' comments:

Reviewer's Responses to Questions

**Comments to the Author**

1. If the authors have adequately addressed your comments raised in a previous round of review and you feel that this manuscript is now acceptable for publication, you may indicate that here to bypass the “Comments to the Author” section, enter your conflict of interest statement in the “Confidential to Editor” section, and submit your "Accept" recommendation.

Reviewer #1: All comments have been addressed

2. Is the manuscript technically sound, and do the data support the conclusions?

Reviewer #1: Yes

3. Has the statistical analysis been performed appropriately and rigorously? 

Reviewer #1: Yes

4. Have the authors made all data underlying the findings in their manuscript fully available?

Reviewer #1: Yes

5. Is the manuscript presented in an intelligible fashion and written in standard English?

Reviewer #1: Yes

6. Review Comments to the Author

Reviewer #1: I want to commend the authors for their openness to addressing the questions that I raised in my initial review and for addressing them in a comprehensive manner.

7. PLOS authors have the option to publish the peer review history of their article (what does this mean?). If published, this will include your full peer review and any attached files.

Reviewer #1: **Yes: **Ademola Adeponle

---

## [Editor Report · Acceptance letter]

12 May 2021

PONE-D-20-36446R1 

Gender differences in suicide-related communication of young suicide victims 

Dear Dr. Balt:

I'm pleased to inform you that your manuscript has been deemed suitable for publication in PLOS ONE. Congratulations! Your manuscript is now with our production department. 

Kind regards, 

on behalf of

Dr. Vincenzo De Luca 

Academic Editor

PLOS ONE